# 'Skeletal Age' for mapping the impact of fracture on mortality

Thach Tran[1,2,3]*, Thao Ho-Le[1], Dana Bliuc[2,3], Bo Abrahamsen[4,5,6], Louise Hansen[7], Peter Vestergaard[8,9,10], Jacqueline R Center[2,3,11], Tuan V Nguyen[1,11,12]*

[1]School of Biomedical Engineering, University of Technology Sydney, Sydney, Australia; [2]Garvan Institute of Medical Research, Sydney, Australia; [3]Faculty of Medicine, UNSW Sydney, New South Wales, Australia; [4]Department of Medicine, Holbæk Hospital, Holbæk, Denmark; [5]Department of Clinical Research, Odense Patient Data Explorative Network, University of Southern Denmark, Odense, Denmark; [6]Nuffield Department of Orthopaedics, Rheumatology and Musculoskeletal Sciences University of Oxford, Oxford, United Kingdom; [7]Kontraktenheden, North Denmark Region, Denmark, Denmark; [8]Department of Clinical Medicine, Aalborg University, Aalborg, Denmark; [9]Department of Endocrinology, Aalborg University Hospital, Aalborg, Denmark; [10]Steno Diabetes Center North Jutland, Aalborg, Denmark; [11]School of Medicine Sydney, University of Notre Dame Australia, Sydney, Australia; [12]School of Population Health, UNSW Medicine, UNSW Sydney, Kensington, Australia

*For correspondence:
SonThach.Tran@uts.edu.au (TT);
TuanVan.Nguyen@uts.edu.au
(TVN)

## Abstract

**Background:** Fragility fracture is associated with an increased risk of mortality, but mortality is not part of doctor-patient communication. Here, we introduce a new concept called 'Skeletal Age' as the age of an individual's skeleton resulting from a fragility fracture to convey the combined risk of fracture and fracture-associated mortality for an individual.

**Methods:** We used the Danish National Hospital Discharge Register which includes the whole-country data of 1,667,339 adults in Denmark born on or before January 1, 1950, who were followed up to December 31, 2016 for incident low-trauma fracture and mortality. Skeletal age is defined as the sum of chronological age and the number of years of life lost (YLL) associated with a fracture. Cox's proportional hazards model was employed to determine the hazard of mortality associated with a specific fracture for a given risk profile, and the hazard was then transformed into YLL using the Gompertz law of mortality.

**Results:** During the median follow-up period of 16 years, there had been 307,870 fractures and 122,744 post-fracture deaths. A fracture was associated with between 1 and 7 years of life lost, with the loss being greater in men than women. Hip fractures incurred the greatest loss of life years. For instance, a 60-year-old individual with a hip fracture is estimated to have a skeletal age of 66 for men and 65 for women. Skeletal Age was estimated for each age and fracture site stratified by gender.

**Conclusions:** We propose 'Skeletal Age' as a new metric to assess the impact of a fragility fracture on an individual's life expectancy. This approach will enhance doctor-patient risk communication about the risks associated with osteoporosis.

**Funding:** National Health and Medical Research Council in Australia and Amgen Competitive Grant Program 2019.

## Editor's evaluation

This important study presents the idea of "Skeletal Age", defined as the age of one's skeleton as a consequence of fragility fracture, as a potential new tool to raise awareness about the increased risk of mortality following a fracture (particularly hip fractures) and thus improve the medical management of osteoporosis. The evidence is convincing and is derived from a very large database from the Danish National Hospital Discharge Registry. The proposed approach is of general interest and might represent a starting point for making the health risks of an osteoporotic fracture more intuitive and possibly more effective.

## Introduction

Fragility fracture is a direct consequence of osteoporosis, just as stroke is a consequence of hypertension. Fracture, especially hip fracture, is associated with an increased risk of mortality. Indeed, patients with a fragility fracture have on average a twofold increase in the risk of mortality (*Center et al., 1999*). Between 22% (*Brauer et al., 2009*) and 58% (*Rapp et al., 2008*) of patients with a hip fracture die within 12 months post fracture. The identification of high-risk individuals for early intervention is a major priority in the control of osteoporotic fractures in the general community. In randomized controlled trials, treating high-risk individuals reduces the risk of fracture (*Black et al., 2020*), though whether the reduction translates into reduced mortality risk remains contentious (*Bolland et al., 2010*; *Cummings et al., 2019*). However, the treatment uptake has been low, with only 40% of hip fracture patients in the United States being given osteoporosis treatment within 1 month after discharge in 2002, which was then halved in 2011 (*Solomon et al., 2014*). This undertreatment is considered a crisis in the management of osteoporosis globally.

Despite the excess mortality after fracture, mortality is not part of doctor-patient communication about treatment or risk assessment, because there is a lack of an intuitive method of conveying risk. Traditionally, relative risk (e.g. "the risk of mortality is increased by twofold") has been used as a metric of excess risk, but this metric often gives an exaggerated impression (*Trevena et al., 2013*), and is perceived as larger than the absolute risk (*Akl et al., 2011*). On the other hand, absolute risk in terms of probability over a period of time is much harder to understand (*Zipkin et al., 2014*), even for doctors and patients (*Gigerenzer et al., 2007*). The underappreciation of post-fracture mortality's gravity has caused patients to be hesitant towards treatment and prevention, contributing to the crisis of osteoporosis management. Thus, there is an urgent need for a more informative metric to internalize the combined risks of fracture and mortality in a way that patients and doctors can comprehend easily.

The concept of 'Effective Age' (*Spiegelhalter, 2016*) is useful here. In engineering, effective age is the age of a structure based on its current conditions; whereas in medicine, the effective age of an individual is the age of a typical healthy person who matches the specific risk profile of this individual (*Spiegelhalter, 2016*). Depending on whether the risk profile is not healthy (i.e. presence of risk factors) or healthy (i.e. absence of risk factors), the effective age is older or younger, respectively than the chronological age (*Spiegelhalter, 2016*). The best-known effective age in medicine is 'Heart Age' or 'Vascular Age' (*NHS Health Check, 2022*) and 'Lung Age' (*Morris and Temple, 1985*). The use of heart age and lung age has resulted in a better clinical impact than the traditional absolute risk metric (*Kulendrarajah et al., 2020*).

We consider that patients with osteoporosis and the general public are not sufficiently informed about the risk of post-fracture mortality, and this might have contributed to the current crisis of under management of osteoporosis (*Roux and Briot, 2020*; *Beaudoin et al., 2019*). In an effort to improve the management of osteoporosis, we advance the idea of the 'Skeletal Age' which is conceptually defined as the age of one's skeleton as a consequence of fragility fracture. This new metric is not only one that quantifies the impact of fracture on mortality but also one that captures the risk of fracture and the risk of post-fracture mortality.

The aim of this study was to analyze the relationship between fracture and mortality, and then translate this relationship into the 'Skeletal Age' for each fracture site by leveraging data from the Danish National Hospital Discharge Registry (NHDR). The NHDR data are ideal for this analysis because, apart from comorbidities at the individual patient level, it has documented the incidence of fractures and post-fracture mortality for the entire Danish population.

**eLife digest** Osteoporosis is a 'silent disease' which often has no immediate symptoms but gradually weakens bones and makes them more likely to break. A bone fracture caused by osteoporosis in people over the age of 50 is linked to long-term health decline and in some cases, even early death. However, poor communication of the mortality risk to patients has led to a low uptake of treatment, resulting in a crisis of osteoporosis management.

The impact of a fracture on life expectancy is typically conveyed to patients and the public in terms of probability (how likely something is to occur) or the relative risk of death compared to other groups. However, statements such as "Your risk of death over the next 10 years is 5% if you have suffered from a bone fracture" can be difficult to comprehend and can lead to patients underestimating the gravity of the risk.

With the aim of devising a new way of conveying risks to patients, Tran et al. analyzed the relationship between fracture and lifespan in over 1.6 million individuals who were 50 years of age or older. The findings showed that one fracture was associated with losing up to 7 years of life, depending on gender, age and fracture site. Based on this finding, Tran et al. proposed the idea of 'skeletal age' as a new metric for quantifying the impact of a fracture on life expectancy.

Skeletal age is the sum of the chronological age of a patient and the estimated number of years of life lost following a fracture. For example, a 60-year-old man with a hip fracture is predicted to lose an estimated 6 years of life, resulting in a skeletal age of around 66. Therefore, this individual has the same life expectancy as a 66-year-old person that has not experienced a fracture.

Skeletal age can also be used to quantify the benefit of osteoporosis treatments. Some approved treatments substantially reduce the likelihood of post-fracture death and translating this into skeletal age could help communicate this to patients. For instance, telling patients that "This treatment will reduce your skeletal age by 2 years" is easier to understand than "This treatment will reduce your risk of death by 25%".

Given the current crisis of osteoporosis management, adopting skeletal age as a new measure of how the skeleton declines after a fracture could enhance doctor-patient communication regarding treatment options and fracture risk assessment. Tran et al. are now developing an online tool called 'BONEcheck.org' to enable health care professionals and the public to calculate skeletal age. Future work should investigate the effectiveness of this new metric in conveying risk to patients, compared with current methods.

## Methods

**Key resources table**

| Reagent type (species) or resource | Designation | Source or reference | Identifiers | Additional information |
|---|---|---|---|---|
| Software, algorithm | R Project for Statistical Computing | R Project for Statistical Computing | RRID:SCR_001905 | |
| Software, algorithm | Stata Statistical Software | A Software resource for statistical analysis and presentation of graphics | RRID:SCR_012763 | |

### Study design

The Danish National Hospital Discharge Registry can be viewed as a retrospective population-based cohort. This analysis included all adults aged 50 years old and older as of January 1, 2001 in Denmark whose health status had been followed up until December 31, 2016 for mortality. Individuals who had sustained a fracture at 45+ years old between 1996 and 2000 were excluded to avoid potential bias that the incident fracture analyzed in this study was a second fracture (*Figure 1*). This is not a clinical trial. This analysis (Statistics Denmark project number 706667) was approved by the National Board of Health, the Danish Data Protection Agency, and Statistics Denmark, and is subject to independent control and monitoring by The Danish Health Data Authority. Written informed consent is waived for routinely collected, pseudonymized registry data. This study followed the Strengthening the Reporting of Observational Studies in Epidemiology (STROBE) reporting guideline.

## Ascertainment of fracture and mortality

The initial incident fracture was defined as the first low-trauma fracture reported between January 1, 2001 and December 31, 2014. When more than one fracture occurred during a single event, only the fracture at the most proximal site was considered. We used the International Statistical Classification of Disease and Related Health Problems, tenth version (ICD-10) codes to identify individuals with specific fracture sites including hip, femur, pelvis, vertebrae, humerus, rib, clavicle (collectively known as proximal fractures), forearm, lower leg, knee, ankle, foot and hand (collectively known as distal fractures) from the Danish NHDR (*Supplementary file 1*). All fractures included in the analysis were radiologically ascertained. Face, skull, finger, or toe fractures and high-trauma fractures due to traffic accidents were excluded.

The study participants were followed up to December 31, 2016 for mortality, allowing at least 2 years of follow-up post fracture. Death was ascertained from the Danish Register on Causes of Death.

## Covariates assessment

The predefined covariates included age and comorbidities. We used the ICD-10 from the NHDR that includes any diagnosis documented between 1996 and 2000 to operationally define comorbidities at the study entry (i.e. January 1, 2001), and those within 5 years prior to the initial fracture to define comorbidities at fracture time. The severity of comorbidities was summarised using the Charlson comorbidity index (*Quan et al., 2011*).

## Statistical analysis

Skeletal age is operationally defined as the sum of the chronological age and the change in life expectancy associated with fracture. The changes in life expectancy resulting from a specific fracture were computed incorporating (i) the association between individual fracture sites and mortality from a Cox's proportional hazards regression and (ii) the baseline hazards described by Gompertz distribution and the population life expectancy from the national lifetable data (*Kulinskaya et al., 2020*).

For the first aim, a Cox's proportional hazards regression was used to quantify the association between an initial specific fracture and mortality, in which both fracture and confounding variables

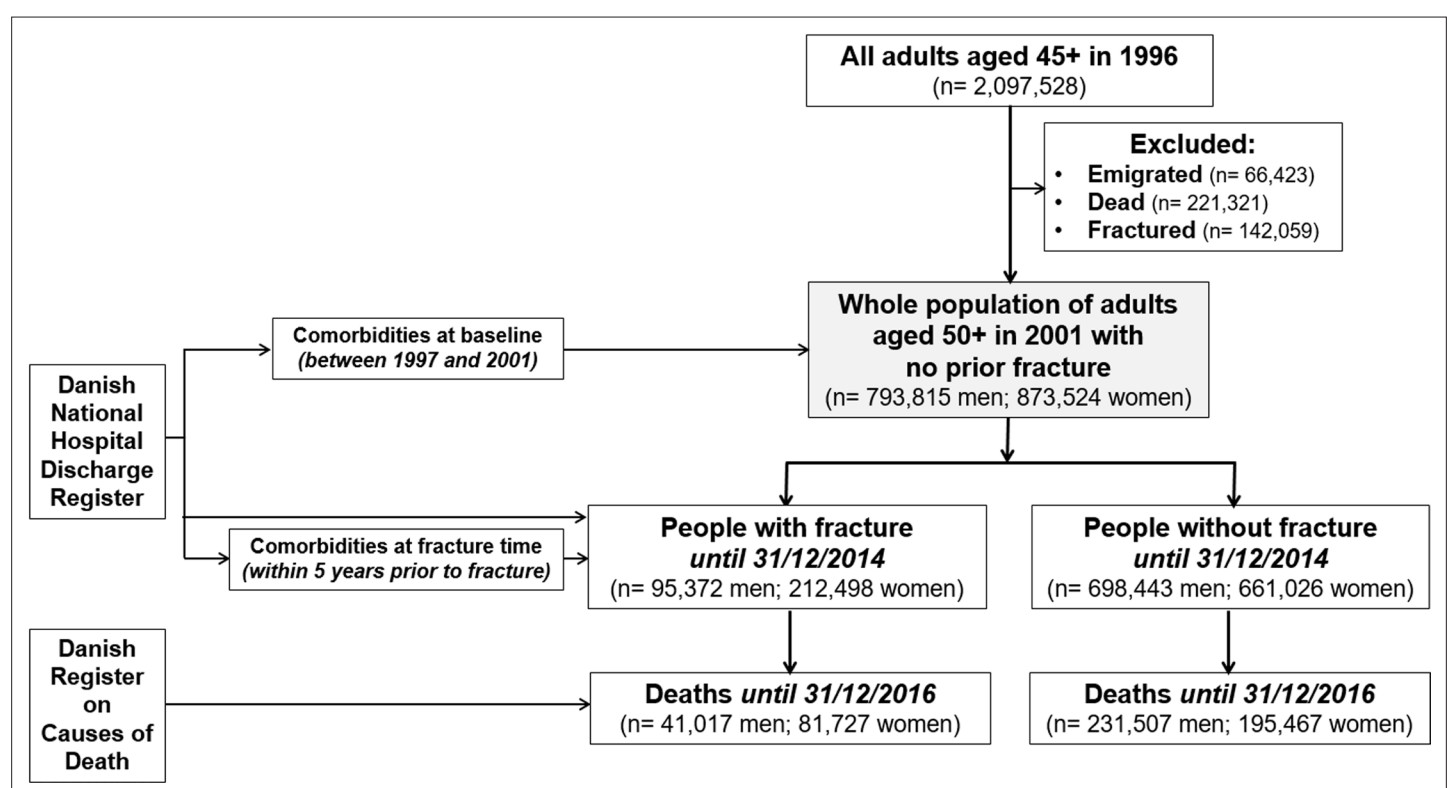

**Figure 1.** Flowchart of recruitment and follow-up.

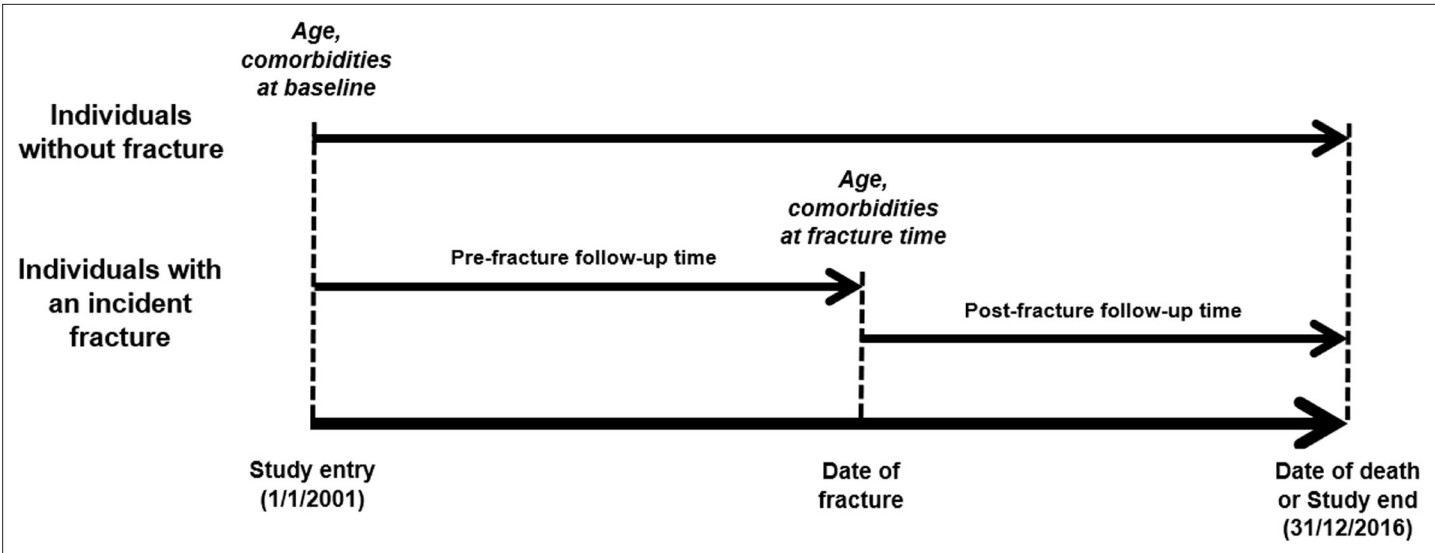

**Figure 2.** Schematic representation of time-dependent analysis.

were analyzed in a time-dependent manner to minimize the risk of immortal time bias (*Suissa, 2008*). The analyses did not account for the possibility of subsequent or recurrent fractures as this falls outside the scope of the study. The models adjusted for confounding effects of age and severity of comorbidities (*Quan et al., 2011*). Age and comorbidities at baseline and those at the time of fracture were also used. For individuals without fracture, the follow-up time was calculated from the study entry to the study end (December 31, 2016) or date of death, whichever came first; while their covariates at baseline were adjusted in the analysis model (*Figure 2*). By contrast, the follow-up time for those with an incident fracture was split into the pre-fracture (i.e. unexposed) and post-fracture (i.e. exposed) period for which the clustering effect was accounted for in the Cox's proportional hazards regression. We conducted a time-dependent analysis to quantify the relationship between an incident exposure (i.e. an incident fracture) and the primary outcome of interest (i.e. post-fracture mortality) accounting for the risk of immortal time bias (*Suissa, 2008*). The pre-fracture period was the time interval between the study entry and date of fracture, and used the covariates at study entry; whereas the post-fracture period was between the date of fracture and date of death or study end, whichever came first, and included the covariates at the time of fracture. The proportional hazards assumption was graphically checked using the Schoenfeld's residuals.

For the second aim, we determined the skeletal age of an individual based on the individual's specific risk profile. First, we calculated a prognostic index for an individual as a single-number summary of the combined effects of his/her specific fracture site and comorbidity severity (*Henderson and Keiding, 2005*). The prognostic index is a linear combination of the risk factors with weights derived from the regression coefficients. The individualized fracture-mortality association for an individual with a specific risk profile was then the difference between his/her prognostic index and the mean prognostic index of 'typical' people in the study population (*Henderson and Keiding, 2005*). In the second step, we used the Gompertz law of mortality and the Danish national lifetable data to transform the fracture-mortality association into life expectancy as a result of a fracture (*Kulinskaya et al., 2020*). The Gompertz law of mortality indicates the annual risk of dying at the age $t$ can be expressed as $h(t) = Be^{kt}$ in which $B \sim 0.0000189$ in women and $0.0000347$ in men (*Gompertz, 1825*). Under the Gompertz law of mortality, the annual risk of dying associated with aging 1 year is remarkably consistent between ages 50 and 95 across ethnicities and over time (*Brenner et al., 1993*; *Vaupel, 2010*). Assuming the Gompertz baseline hazard, the log-hazards of mortality to specific fracture site $i$ at the fracture time $t$ is $a + bt$, where $a$ and $b$ are the estimated regression coefficients. The life expectancy for a specific fracture site at the fracture age $z$ under the Gompertz distribution $G(a,b)$ is obtained as $e_{G(a,b)}(z) = \dfrac{b^{-1}\exp(b^{-1}e^{a})E_{1}(b^{-1}e^{a+bz})}{\exp(-e^{a}b^{-1}(e^{bz}-1))}$ , where $E_{1}(b^{-1}e^{a+bz})$ denotes the exponential integral. The

loss of life years associated with specific fracture site was then calculated as the difference between the estimate life expectancy associated with fracture $e_{G(a,b)}(z)$ and the population life expectancy (*Kulinskaya et al., 2020*). The skeletal age for each individual site of fracture was the sum of the individual's chronological age at the time of fracture and the loss of life years as a result of the fracture. The 95% CI of the skeletal age was computed using the confidence limits of the adjusted hazards ratios to account for uncertainty of the magnitude of association between individual site of fracture and mortality. The analyses were performed using Stata (*Stata Statistical Software: Release 16*. College Station, TX: StataCorp LLC) and the R statistical environment on a Windows platform (*R Development Core Team, 2020*).

## Results

### Incidence of fractures

The present analysis was based on data from 793,815 men and 873,524 women aged 63.9 (standard deviation [SD] 10.3) and 65.5 (11.3) years old as of January 1, 2001, respectively (*Figure 1*). During a median of 14 years of follow-up (interquartile range [IQR]: 6.5, 14.0 years), 95,372 men and 212,498 women had sustained a fracture. The incidence of fractures was 10.9 fractures/1000 person-years (95% confidence interval [CI]: 10.8, 11.0) in men and 23.2 fractures/1000 person-years (23.1, 23.3) in women. Collectively, forearm, hip, and humerus fractures accounted for 55% of all fractures in men and 70% in women.

As expected, men and women with a fracture were on average older than those without a fracture. Individuals with a fracture had more comorbidities than those who did not sustain a fracture (*Table 1*). For instance, the prevalence of myocardial infarction among patients with a fracture (9.3% in men and 4.7% in women) was higher than that among individuals without a fracture (4% in men and 1.5% in women). Similar trend was also observed for stroke, diabetes, and cancer.

### Incidence of mortality

During a median follow-up of 6.5 years (6.0 years (IQR: 2.5, 10.0) in men, 6.7 years (3.3, 10.8) in women), 272,524 men and 277,194 women had died, yielding unadjusted mortality incidence rates of 6.55 deaths/100 person-years (95% CI: 6.48, 6.61) in men and 5.42 deaths/100 person-years (5.39, 5.46) in women. More importantly, the unadjusted rate of mortality among patients with a fracture (6.54/100 person-years in men and 5.42/100 person-years in women) was greater than among those without a fracture (2.56/100 person-years in men and 2.24/100 person-years in women). Analysis by fracture site revealed that men and women with a fracture at the hip, pelvis, or vertebra had a much greater risk of mortality than other fractures (*Table 2*).

However, the above observation could be confounded by age and comorbidities, because individuals with a fracture were on average older and had more comorbidities than those without a fracture. Therefore, we employed multivariable-adjusted Cox's proportional hazards model to estimate the strength of the association between fracture and mortality, adjusting for age and severity of comorbidities (*Figure 3*). Individuals with any fragility fracture were associated with a 30–45% increased hazard of death (adjusted hazard ratio = 1.46 (95% CI: 1.45, 1.48) in men; 1.28 (1.27, 1.30) in women). For the same age and comorbidity profile, men and women with a hip or femur fracture had an almost twofold greater risk of death than those without a fracture.

The increased risk of mortality was also observed among those with a proximal fracture, such as the pelvis, vertebrae, humerus, rib, clavicle, and lower leg fracture after controlling for age and comorbidities. However, there was no significantly increased risk of death following a forearm, knee, ankle, hand, or foot fracture (*Figure 3*).

### Skeletal age

Based on the association between fracture and mortality and using the Gompertz law, we estimated the skeletal age for each fracture site and each chronological age (from the age of 50) in men and women (*Figure 4*; *Supplementary file 2*). Patients with a fracture - any fracture at all - have lost years of life, and hence their skeletal age was greater than their chronological age. For a given age, men with a fracture on average had greater skeletal age than women by approximately 1 year. In either

**Table 1.** Characteristics of the study population at baseline stratified by gender, fracture, and mortality status.

| Characteristic | Non-fracture group | | Fracture group | |
| --- | --- | --- | --- | --- |
| | **Alive** | **Dead** | **Alive** | **Dead** |
| **Men** | | | | |
| Number of individuals | 466,936 | 231,507 | 54,355 | 41,017 |
| Age at baseline[1] | 59.8 (7.7) | 71.1 (10.4) | 61.3 (8.9) | 72.4 (10.4) |
| Age at fracture[1] | | | 68.2 (9.9) | 77.8 (10.5) |
| Comorbidities* | | | | |
| *Charlson comorbidity index*[2] | 0 (0, 0) | 0 (0, 1) | 0 (0, 1) | 1 (0, 2) |
| 0 | 404,415 (86.7%) | 140,293 (60.6%) | 32,287 (59.4%) | 12,592 (30.7%) |
| 1–2 | 56,701 (12.1%) | 70,841 (30.6%) | 17,557 (32.3%) | 19,073 (46.5%) |
| 3–4 | 3,328 (0.7%) | 9,260 (4.0%) | 2,881 (5.3%) | 5,209 (12.7%) |
| 5+ | 2,492 (0.5%) | 1,112 (4.8%) | 1,631 (3.0%) | 4,143 (10.1%) |
| *Specific comorbidities*[3] | | | | |
| Myocardial infarction | 18,734 (4.0%) | 22,225 (9.6%) | 5,055 (9.3%) | 6,071 (14.8%) |
| Congestive heart failure | 12,814 (2.7%) | 26,623 (11.5%) | 5,109 (9.4%) | 8,655 (21.1%) |
| Stroke | 21,792 (4.7%) | 31,253 (13.5%) | 9,458 (17.4%) | 11,567 (28.2%) |
| Peripheral vascular disease | 10,912 (2.3%) | 18,058 (7.8%) | 4,240 (7.8%) | 5,660 (13.8%) |
| Arrhythmias | 10,835 (2.3%) | 18,289 (7.9%) | 4,892 (9.0%) | 5,747 (18.4%) |
| Hypertension | 14,340 (3.1%) | 15,279 (6.6%) | 7,718 (14.2%) | 7,875 (19.2%) |
| Diabetes mellitus with no chronic complications | 19,805 (4.2%) | 21,067 (9.1%) | 6,305 (11.6%) | 6,317 (15.4%) |
| Diabetes mellitus with chronic complications | 6,120 (1.3%) | 8,103 (3.5%) | 2,446 (4.5%) | 2,748 (6.7%) |
| Renal disease | 4,290 (0.9%) | 8,797 (3.8%) | 2,120 (3.9%) | 3,609 (8.8%) |
| Cancer | 26,126 (5.6%) | 43,523 (18.8%) | 8,914 (16.4%) | 11,813 (28.8%) |
| Metastatic tumors | 1,894 (0.4%) | 9,492 (4.1%) | 1,033 (1.9%) | 3,035 (7.4%) |
| Dementia | 2,869 (0.6%) | 7,640 (3.3%) | 2,663 (4.9%) | 6,071 (14.8%) |
| Chronic lung disease | 16,600 (3.6%) | 25,697 (11.1%) | 6,740 (12.4%) | 8,696 (21.2%) |
| Rheumatologic | 4,881 (1.1%) | 4,630 (2.0%) | 1,685 (3.1%) | 1,600 (3.9%) |
| Mild liver disease | 2,647 (0.6%) | 3,704 (1.6%) | 1,468 (2.7%) | 1,805 (4.4%) |
| Moderate/severe liver disease | 589 (0.1%) | 1,389 (0.6%) | 435 (0.8%) | 779 (1.9%) |
| Hemiplegia | 661 (0.1%) | 927 (0.4%) | 461 (0.9%) | 480 (0.9%) |
| HIV/AIDS | 108 (0.02%) | 81 (0.03%) | 37 (0.07%) | 23 (0.06%) |
| **Women** | | | | |
| Number of individuals | 465,559 | 195,467 | 130,771 | 81,727 |
| Age at baseline[1] | 60.5 (8.3) | 74.0 (11.1) | 63.9 (9.7) | 76.3 (9.9) |
| Age at fracture[1] | | | 70.9 (10.2) | 81.3 (9.6) |

*Table 1 continued on next page*

*Table 1 continued*

| Characteristic | Non-fracture group | | Fracture group | |
|---|---|---|---|---|
| | **Alive** | **Dead** | **Alive** | **Dead** |
| Comorbidities* | | | | |
| *Charlson comorbidity index*[2] | 0 (0, 0) | 0 (0, 1) | 0 (0, 1) | 1 (0, 2) |
| 0 | 400,005 (85.9%) | 121,776 (62.3%) | 81,574 (62.4%) | 32,093 (39.3%) |
| 1–2 | 59,923 (12.9%) | 57,663 (29.5%) | 41,114 (31.4%) | 35,866 (44.0%) |
| 3–4 | 2,975 (0.6%) | 6,255 (3.2%) | 5,196 (4.0%) | 7,960 (9.8%) |
| 5+ | 2,656 (0.6%) | 9,773 (5.0%) | 2,887 (2.2%) | 5,695 (7.0%) |
| *Specific comorbidities*[3] | | | | |
| Myocardial infarction | 7,164 (1.5%) | 10,946 (5.6%) | 6,109 (4.7%) | 7,562 (9.3%) |
| Congestive heart failure | 7,307 (1.6%) | 18,374 (9.4%) | 8,056 (6.2%) | 13,615 (16.7%) |
| Stroke | 16,329 (3.5%) | 22,870 (11.7%) | 17,143 (13.1%) | 19,247 (23.6%) |
| Peripheral vascular disease | 6,517 (1.4%) | 10,555 (5.4%) | 6,196 (4.7%) | 7,309 (9.0%) |
| Arrhythmias | 7,671 (1.7%) | 13,096 (6.7%) | 7,947 (6.1%) | 11,440 (14.0%) |
| Hypertension | 12,910 (2.8%) | 12,510 (6.4%) | 17,970 (13.7%) | 15,727 (19.3%) |
| Diabetes mellitus with no chronic complications | 14,151 (3.0%) | 14,074 (7.2%) | 9,960 (7.6%) | 9,214 (11.3%) |
| Diabetes mellitus with chronic complications | 3,367 (0.7%) | 4,496 (2.3%) | 2,969 (2.3%) | 3,126 (3.8%) |
| Renal disease | 2,386 (0.5%) | 4,300 (2.2%) | 2,697 (2.1%) | 3,561 (4.4%) |
| Cancer | 30,913 (6.6%) | 34,989 (17.9%) | 20,246 (15.5%) | 18,785 (23.0%) |
| Metastatic tumors | 2,302 (0.5%) | 8,796 (4.5%) | 2,221 (1.7%) | 4,632 (5.7%) |
| Dementia | 3,035 (0.7%) | 7,428 (3.8%) | 7,169 (5.5%) | 13,136 (16.1%) |
| Chronic lung disease | 18,759 (4.0%) | 20,915 (10.7%) | 15,065 (11.5%) | 14,018 (17.2%) |
| Rheumatologic | 10,392 (2.2%) | 7,037 (3.6%) | 7,482 (5.7%) | 5,256 (6.4%) |
| Mild liver disease | 2,599 (0.6%) | 2,346 (1.2%) | 2,165 (1.7%) | 1,802 (2.2%) |
| Moderate/severe liver disease | 296 (0.1%) | 782 (0.4%) | 422 (0.3%) | 643 (0.8%) |
| Hemiplegia | 524 (0.1%) | 573 (0.3%) | 577 (0.4%) | 475 (0.6%) |
| HIV/AIDS | 14 (0.0%) | 9 (0.0%) | 12 (0.01%) | 4 (0.0%) |

Notes: [1]: mean (SD); [2]: median (IQR); [3]: number (%). *: at baseline for individuals without fracture or at fracture time for fracture patients. Comorbidities included not only chronic diseases that require hospitalization, but also those documented as either secondary diagnoses or at outpatient or emergency visits.

sex, the loss of years of life was more pronounced in younger age groups and gradually converged in the older age groups.

As expected, patients with a hip fracture had the highest skeletal age than those with other fractures. For example, a 70-year-old man who had sustained a hip fracture would have a skeletal age of 75 years (i.e. a loss of 5 years of life); however, if a 50-year-old man with a hip fracture would have a skeletal age of 56.8 years which is equivalent to almost 7 years of life lost. Other fractures such as the femur, pelvis, vertebrae, and humerus also signified a significant loss of years of life (around 5 years);

**Table 2.** Incidence of mortality following specific fracture sites stratified by gender.

| Fracture | Number | Age at fracture (years) | Number of deaths | Follow-up time (person-years) | Rate* of mortality (95% CI) |
|---|---|---|---|---|---|
| **Men** | | | | | |
| No fracture | 698,443 | 63.6** (10.2) | 231,507 | 9,049,194 | 2.6% (2.5, 2.6) |
| Any fracture | 95,372 | 72.3 (11.2) | 41,017 | 626,733 | 6.5% (6.5, 6.6) |
| Hip fracture | 25,706 | 79.5 (9.7) | 16,890 | 107,789 | 15.7% (15.4, 15.9) |
| Femur | 1,910 | 74.5 (10.8) | 997 | 10,468 | 9.5% (8.9, 10.1) |
| Pelvis | 1,305 | 77.0 (10.9) | 751 | 6,465 | 11.6% (10.8, 12.5) |
| Vertebrae | 6,924 | 72.7 (10.5) | 3,125 | 41,746 | 7.5% (7.2, 7.7) |
| Humerus | 10,126 | 72.6 (10.7) | 4,827 | 62,350 | 7.7% (7.5, 8.0) |
| Rib | 6,867 | 69.7 (10.4) | 2,350 | 50,186 | 4.7% (4.5, 4.9) |
| Clavicle | 5,201 | 68.2 (10.5) | 1,636 | 39,731 | 4.1% (3.9, 4.3) |
| Lower leg | 6,296 | 67.0 (9.5) | 1,796 | 52,915 | 3.4% (3.2, 3.6) |
| Forearm | 15,268 | 69.4 (10.2) | 4,675 | 120,719 | 3.9% (3.8, 4.0) |
| Knee | 1,382 | 70.2 (10.1) | 447 | 10,886 | 4.1% (3.7, 4.5 |
| Ankle | 1,084 | 67.6 (9.8) | 275 | 9,081 | 3.0% (2.7, 3.4) |
| Hand | 8,309 | 67.7 (10.1) | 2,178 | 70,263 | 3.1% (3.0, 3.2) |
| Foot | 4,994 | 65.3 (9.7) | 1,070 | 44,136 | 2.4% (2.3, 2.6) |
| **Women** | | | | | |
| No fracture | 661,026 | 64.5** (11.1) | 195,467 | 8,723,863 | 2.2% (2.2, 2.3) |
| Any fracture | 212,498 | 74.9 (11.2) | 81,727 | 1,506,940 | 5.4% (5.4, 5.5) |
| Hip fracture | 51,669 | 82.1 (9.2) | 31,816 | 259,087 | 12.3% (12.1, 12.4) |
| Femur | 3,529 | 78.9 (10.9) | 1,921 | 19,322 | 9.9% (9.5, 10.4) |
| Pelvis | 4,920 | 81.2 (10.0) | 2,893 | 26,761 | 10.8% (10.4, 11.2) |
| Vertebrae | 9,732 | 76.6 (10.6) | 4,658 | 60,295 | 7.7% (7.5, 7.9) |
| Humerus | 28,298 | 74.4 (10.5) | 10,330 | 202,662 | 5.1% (5.0, 5.2) |
| Rib | 3,153 | 74.1 (11.9) | 1,218 | 22,285 | 5.5% (5.2, 5.8) |
| Clavicle | 4,480 | 72.7 (11.5) | 1,577 | 31,755 | 5.0% (4.7, 5.2) |
| Lower leg | 11,460 | 70.1 (10.8) | 3,130 | 94,308 | 3.3% (3.2, 3.4) |
| Forearm | 68,333 | 72.0 (10.3) | 18,470 | 559,406 | 3.3% (3.2, 3.3) |
| Knee | 2,604 | 71.3 (9.7) | 629 | 21,085 | 3.0% (2.8, 3.2) |
| Ankle | 1,936 | 70.0 (10.6) | 499 | 15,726 | 3.2% (2.9, 3.5) |
| Hand | 12,489 | 69.6 (10.2) | 2681 | 108,112 | 2.5% (2.4, 2.6) |
| Foot | 9,916 | 67.8 (9.6) | 1904 | 86,136 | 2.2% (2.1, 2.3) |

Notes: *: rates were calculated as number of deaths/100 person-years; **: age at baseline (years).

thus, 50-year-old patients with one of these fractures are estimated to have a skeletal age of around 55 years.

However, fractures are the rib, clavicle, and lower leg were associated with lower years of life lost, and the skeletal age of patients with one of these fractures was generally lower than patients with a more serious fracture (e.g. hip fracture). For instance, a 60-year-old patient with a lower leg fracture

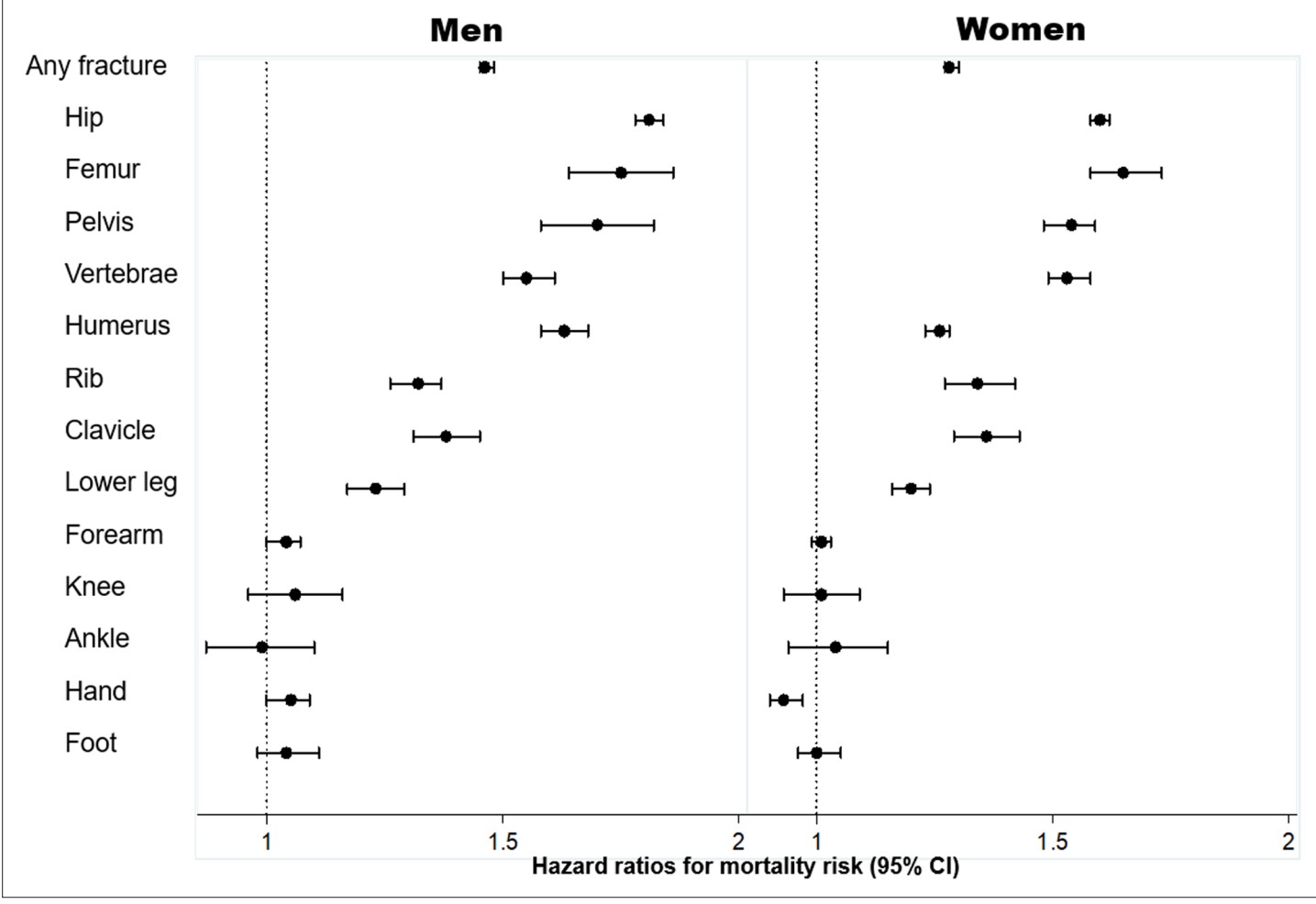

**Figure 3.** Association between specific fracture and mortality risk, adjusted for age and severity of comorbidities*: hazard ratio and 95% confidence interval (CI). Legend * severity of comorbidities is assessed using the Charlson Comorbidity Index. Hazard ratio and the corresponding 95% confidence interval were estimated from a multivariable-adjusted Cox's proportional hazards regression.

would be expected to have a skeletal age of 62.4 years for men or 61.9 years for women (*Supplementary file 3*).

## Discussion

It has been well established that fracture, especially hip fracture, is associated with an increased risk of mortality, and this excess risk is commonly expressed in terms of the relative risk metric. Here, we proposed a new effective age metric called 'Skeletal Age' to quantify the impact of fracture on mortality. Using data from a nationwide cohort of 1.7 million adults aged 50+ years old in Denmark, we showed that almost all types of fracture were associated with a loss of years of life, indicating that the skeletal age of individuals suffering from a fracture is greater than their chronological age. This finding has important implications that we discuss below.

Our finding confirmed the previous studies (*Tran et al., 2018*; *Bliuc et al., 2009*; *Browner et al., 1996*; *Melton et al., 2013*; *Haentjens et al., 2010*; *Tran et al., 2017*) that patients with a fragility fracture were associated with a significantly greater risk of mortality than their similarly aged and gender counterparts without fracture who had the same comorbidity profile. Our finding is also consistent with an earlier Danish study demonstrating reduced life expectancy in patients, with or without fractures, at the time of beginning osteoporosis treatment (*Abrahamsen et al., 2015*). However, the magnitude of the association between individual sites of fracture observed in our study is slightly lower than that documented in several previous cohort studies (*Bliuc et al., 2009*; *Browner et al., 1996*; *Melton*

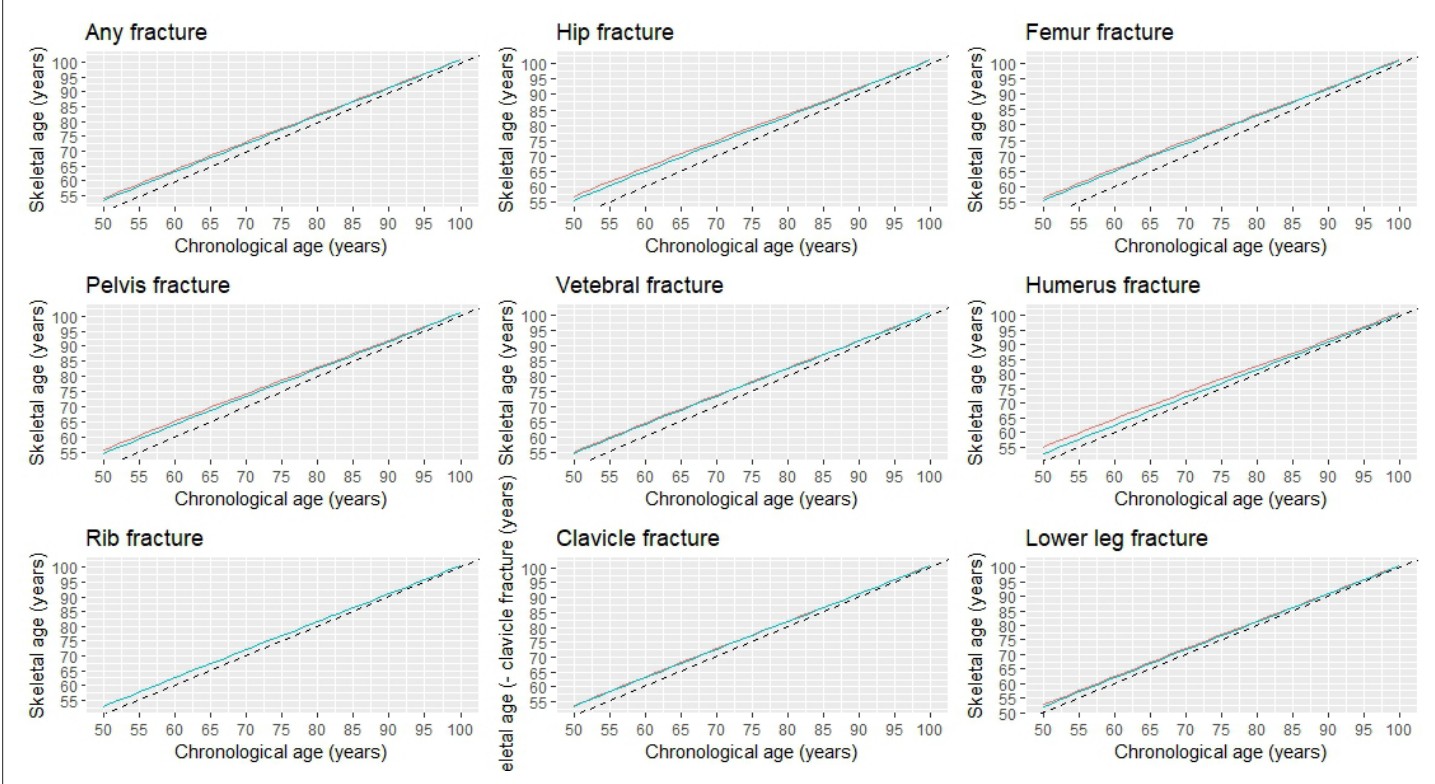

**Figure 4.** Skeletal age by specific fracture site and chronological age at fracture.

*et al., 2013*; *Haentjens et al., 2010*; *Tran et al., 2017*), with hip fractures being associated with a 1.7-to-6-fold increased risk of death (*Haentjens et al., 2010*). This discrepancy could be due to the difference in study populations. While cohort studies usually recruit healthy participants, our national registry-based study was able to document all individuals in Denmark. As a result, our control group (i.e. those without a fracture) included individuals in poor health conditions with multiple comorbidities who are usually unable to be recruited in a cohort study. Whether the association between fracture and mortality is causal or not is a matter of contention. It is commonly assumed that the association is confounded by comorbidities, but multiple studies have found that comorbidities contributed little to the excess risk of mortality among fractured patients (*Vestergaard et al., 2007*; *Chen et al., 2018*). In the present analysis, we have also adjusted for comorbidities, and the significant fracture – mortality association remained unchanged, suggesting that the association is unlikely due to comorbidities.

Regardless of the nature of the association, mortality is not a component of doctor-patient communication. Existing fracture risk assessment models such as Garvan (*Nguyen et al., 2007*) and FRAX (*Kanis et al., 2007*) provide the estimated probability (i.e. absolute risk) of fracture over a 10 year period without any information on mortality. However, doctors and patients have difficulty in understanding and interpreting probability (*Gigerenzer et al., 2007*). Only a fifth of a sample of highly educated American adults could understand one in 1000 is equivalent to 0.1% (*Lipkus et al., 2001*). Conveying a low, though clinically significant absolute risk (e.g. '*Your risk of hip fracture over the next 10 years is 5%*') might provide a false peace of mind and underestimated risk perception, leading to increasing possibilities of refusing the recommended treatment (*Trevena et al., 2013*). Thus, poor communication about fracture risk and mortality consequences might have contributed to the global crisis of undertreatment of osteoporosis.

Based on the concept of effective age, we proposed the idea of 'Skeletal Age' as a new metric for communicating the risk of mortality following a fracture. Unlike the current fracture risk assessment tools (*Beaudoin et al., 2019*) which estimate the probability of fracture over time using probability-based metrics, such as relative risk and absolute risk, skeletal age quantifies the consequence of a fracture using a natural frequency metric. A natural frequency metric has been consistently shown to be easier and more friendly to doctors and patients than the probability-based metrics (*Akl et al., 2011*;

*Gigerenzer et al., 2007*; *Ahmed et al., 2012*). It is not straightforward to appreciate the importance of the twofold increased risk of death (i.e. relative risk = 2.0) without knowing the background risk (i.e. twofolds of 1% would remarkably differ from twofolds of 10%). By contrast, for the same twofold mortality risk of hip fracture, telling a 60 year man with a hip fracture that his skeletal age would be 66 years old, equivalent to a 6 year loss of life, is more intuitive. The skeletal age of 66 years old can also be interpreted as the individual being in the same risk category as a 66-year-old with 'favorable risk factors' or at least the ones that are potentially modifiable. Hence, an older skeletal age also means a greater risk of fracture. In addition, the skeletal age can be also used to convey the possible benefit of treatment, providing an alternative metric to the conventional metrics such as relative risk or relative risk reduction. For instance, a patient might find the statement '*Zoledronic acid treatment helps a patient with a hip fracture gain 3 years of life*' much easier to understand and probably more persuasive than '*Zoledronic acid treatment reduced the risk of death by 28%*' (*Lyles et al., 2007*).

Skeletal age is an addition to the already available metrics such as 'Heart Age,' 'Lung Age,' and more recently 'Covid Age' which have been demonstrated to be superior to the conventional risk metrics (*Grover et al., 2007*; *Lowensteyn et al., 1998*; *Lopez-Gonzalez et al., 2015*; *Bonner et al., 2015*; *Svendsen et al., 2020*) in terms of positive behavioral changes. For instance, compared with usual care, individuals who were given heart age had a greater smoking cessation rate (*Lopez-Gonzalez et al., 2015*; *Bonner et al., 2015*), substantial improvement in weight, body mass index, and physical activity (*Lopez-Gonzalez et al., 2015*), and a greater proportion of high-risk patients returning for a follow-up appointment (*Lowensteyn et al., 1998*). The use of heart age also led to significantly greater improvement at 12 months in metabolic parameters (*Lopez-Gonzalez et al., 2015*) or lipid profiles (*Grover et al., 2007*). A recent cluster randomized controlled trial in Norway also found that heart age was a good way to communicate cardiovascular risk and to motivate individuals to reduce cardiovascular risk factors (*Svendsen et al., 2020*). Similarly, the use of lung age could also lead to a clinically significant increase in the reported smoking cessation rates (*Kaminsky et al., 2011*; *Takagi et al., 2017*; *Parkes et al., 2008*). Collectively, these data suggest that effective age metrics can help patients better understand their risk and lead to preventive changes. A web-based calculator 'BONEcheck.org' has been developed to assist healthcare professionals and the public to estimate the risk of fracture and skeletal age for an individual with specific risk profiles, potentially improving doctor-patient risk communication.

Our findings should be interpreted within the context of their strengths and limitations. First, we included a whole-country population with long follow-up and robust diagnostic data, minimizing potential selection bias and misclassification (*Frank, 2000*), and allowing us to examine mortality risk following individual fracture sites. Second, the current study analyzed both fracture and covariates, such as aging and the presence of comorbidities in a time-dependent manner, making it statistically rigorous in minimizing an immortal time bias and sufficiently accounting for confounding effects. The analysis thus provided accurate estimates of the association between specific fracture sites and mortality risk as it was able to control for confounding effects at both the study entry and the time of fracture.

However, the use of registry-based data, originally documented and coded for administrative or reimbursement purposes is prone to variable data accuracy, the lack of specific clinical information, and non-medical factors (*Mazzali and Duca, 2015*). Fortunately, all essential study variables (i.e. comorbidities, fracture, and death) were systematically obtained from the Danish NHDR that includes excellent, complete medical records and precise diagnoses for all individuals living in Denmark since 1995 (*Andersen et al., 1999*; *Vestergaard and Mosekilde, 2002*). The use of diagnosis-based comorbidities, though possibly associated with lower sensitivity than the self-reported ones (*Lujic et al., 2017*) is able to capture the severe health disorders that prompt the participants to seek for medical assistance. Second, the analyses could not make adjustments for lifestyle factors and physical activity, which are usually obtained in a cohort study. As they are closely related to aging and the presence of comorbidities which were already accounted for in our analysis, making further adjustments for these lifestyle factors is unlikely to substantially modify the findings. Third, mild, asymptomatic fractures or chronic diseases that did not require medical attention might not be registered in the Danish NHDR. It is nevertheless unlikely that these unregistered fractures and comorbidities would modify the findings significantly, as previous studies have indicated a high concordance between self-reported fractures and those registered in the NHDR (*Hundrup et al., 2004*), and a high positive

predictive value of the self-reported comorbidities (*Lujic et al., 2017*). Finally, our analysis did not include the examination of the effect of bone metabolism-affecting medications on the assessment of skeletal health. However, these medications were prescribed to only a few individuals aged 50 years or over in Denmark (*Vestergaard et al., 2005*), even for those with a hip fracture (*Roerholt et al., 2009*), and there is no strong evidence from randomized controlled trials of their impact on post-fracture mortality (*Cummings et al., 2019*).

In conclusion, we advanced the concept of skeletal age as the age of an individual's skeleton resulting from a fragility fracture. Unlike existing metrics (e.g. relative risk, probability of fracture), skeletal age combines the risk that an individual will sustain a fracture and the risk of mortality once a fracture has occurred, making the doctor-patient communication more intuitive and possibly more effective. Given the evidence of the successful implication of similar effective age metrics, skeletal age is expected to improve risk communication and ultimately improve treatment uptake among patients who are indicated for treatment. A randomized controlled trial aiming to compare the use of skeletal age and the current metrics in fracture risk communication is warranted.

# Additional information

## Competing interests

Bo Abrahamsen: has received institutional research grants from UCB, Kyowa-Kirin and Pharmacosmos; consulting fees from UCB and Kyowa-Kirin; and fees for lectures from Amgen, Gedeon-Richter, Kyowa-Kirin, Eli Lily and Pharmacosmos. The author is also on the Advisory Board for UCB and is president of European Calcified Tissue Society. The author has no other competing interests to declare. Jacqueline R Center: has received honoraria for educational talks and Advisory boards from Amgen and honoraria for an Advisory board from Bayer. The author has no other competing interests to declare. Tuan V Nguyen: has received grants from Australian National Health and Medical Research Council and Amgen Competitive Grant Program; fees for lectures from Amgen, Bridge Health Care (VN), DKSH Pharma, MSD and VT Health Care (VN); and support from Amgen for attending the annual meeting of APCO and from VT Health Care (VN) for attending the Vietnam Osteoporosis Society Annual Scientific Meeting. The author is also Chair of the Research Committee for Australian and New Zealand Bone and Mineral Society, a Senior Advisor for Vietnam Osteoporosis Society, and an Executive Member of Asia Pacific Consortium on Osteoporosis. The author has no other competing interests to declare. The other authors declare that no competing interests exist.

## Funding

| Funder | Grant reference number | Author |
|---|---|---|
| National Health and Medical Research Council | APP1195305 | Tuan V Nguyen |
| Amgen Competing Grant Program | | Tuan V Nguyen |

The funders had no role in study design, data collection and interpretation, or the decision to submit the work for publication.

## Author contributions

Thach Tran, Conceptualization, Data curation, Software, Formal analysis, Validation, Visualization, Methodology, Writing – original draft, Writing – review and editing; Thao Ho-Le, Conceptualization, Formal analysis, Methodology, Writing – review and editing; Dana Bliuc, Bo Abrahamsen, Louise Hansen, Peter Vestergaard, Jacqueline R Center, Data curation, Writing – review and editing; Tuan V Nguyen, Conceptualization, Data curation, Software, Formal analysis, Supervision, Funding acquisition, Visualization, Methodology, Writing – original draft, Project administration, Writing – review and editing

## Author ORCIDs

Thach Tran ⓘ http://orcid.org/0000-0002-6454-124X
Louise Hansen ⓘ http://orcid.org/0000-0002-4400-2732

Jacqueline R Center http://orcid.org/0000-0002-5278-4527
Tuan V Nguyen http://orcid.org/0000-0002-3246-6281

### Ethics

Human subjects: This analysis (Statistics Denmark project number 706667) was approved by the National Board of Health, the Danish Data Protection Agency, and Statistics Denmark, and subject to independent control and monitoring by The Danish Health Data Authority. Written informed consent is waived for routinely collected, pseudonymized registry data.

### Decision letter and Author response

Decision letter https://doi.org/10.7554/eLife.83888.sa1
Author response https://doi.org/10.7554/eLife.83888.sa2

## Additional files

### Supplementary files

- Supplementary file 1. List of ICD-10 codes used to define specific fractures and comorbidities.
- Supplementary file 2. Skeletal age by specific fracture site and chronological age at fracture.
- Supplementary file 3. Skeletal age for a 60-year-old individual who sustained a fracture at a specific bone.
- Supplementary file 4. R codes used to construct skeletal age for individual fracture sites associated with increased mortality risk.
- MDAR checklist

### Data availability

The study used data from the Danish National Patient Register which individual patient data cannot be shared without authorised access, as per Danish Legislation (https://www.dst.dk/en/TilSalg/Forskningsservice/Dataadgang). To be able to access data interested researchers need to be employees of Danish university, university hospital or other specified Danish organizations (https://www.dst.dk/ext/594574631/0/forskning/Regler-for-adgang-til-pseudomyiserede-mikrodata-under-Danmarks-Statistiks-forskerordninger--pdf). Non-Danish citizens may obtain access if employed at said institutions. The interested researchers will need to be registered as an authorized research group under Statistics Denmark (https://www.dst.dk/da/TilSalg/Forskningsservice/Dataadgang/Autorisering) and fulfill the criteria mentioned above. The research proposal expected to present the specific aims and the feasibility is examined and approved by the Danish Data Protection Agency and the Danish Health Data Authority [https://sundhedsdatastyrelsen.dk/da/forskerservice/ansog-om-data]. However, commercial research cannot be performed (https://www.dst.dk/ext/594574631/0/forskning/Regler-for-adgang-til-pseudomyiserede-mikrodata-under-Danmarks-Statistiks-forskerordninger--pdf). Deidentified data may not be removed from the server (https://www.dst.dk/da/TilSalg/Forskningsservice/hjemtagelse-af-analyseresultater). Aggregated data can only be presented if no individuals can be identified (e.g., counts in a contingency table must not be below five). Supplementary File 1 provides the ICD-10 codes from the Danish National Patient Discharge Register used to define fractures and comorbidities in this analysis. The aggregated data and the R analysis codes utilized to estimate skeletal age for each individual fracture site and to generate the graphs and charts in the manuscript (Skeletal-Age.html), and the analysis output (Skeletal Age_Rmarkdown.pdf) are publicly accessible (https://github.com/ThachSTran/Skeletal-Age.git, copy archived at swh:1:rev:db2c39ebf11f55612a745140ea078d7afa279bb0).

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
