## [Editor Report]

This important study presents the idea of "Skeletal Age", defined as the age of one's skeleton as a consequence of fragility fracture, as a potential new tool to raise awareness about the increased risk of mortality following a fracture (particularly hip fractures) and thus improve the medical management of osteoporosis. The evidence is convincing and is derived from a very large database from the Danish National Hospital Discharge Registry. The proposed approach is of general interest and might represent a starting point for making the health risks of an osteoporotic fracture more intuitive and possibly more effective.

---

## [Decision Letter]

**Decision letter after peer review:**

Thank you for submitting your article "'Skeletal Age' as a measure of the impact of fracture on mortality: a nationwide population-based study" for consideration by *eLife*. Your article has been reviewed by 2 peer reviewers, and the evaluation has been overseen by a Reviewing Editor and Mone Zaidi as the Senior Editor. The following individual involved in the review of your submission has agreed to reveal their identity: Elena Tsourdi (Reviewer #2).

Essential revisions:

1) Better clarification of study design and statistical analysis should be required in order to address the conceptual problems raised by Reviewer 1 (e.g. concerning the assessment of comorbidities, or how the information about low-energy fracture obtained from ICD registries, or whether the risk of a second fracture/recurrent fractures was taken into account in the analyses). Additional information is also required about the identification of fractures, particularly vertebral fractures, as outlined by Reviewer 2 (e.g. were all fractures radiographically confirmed?).

2) As underlined by Reviewer 1, the analysis performed did not show that individuals with and without fractures matched for co-morbidities had different mortality risks. Given the large and comprehensive sample, this could be explored in more detail.

3) Additional information is required to better explain how the assessment of skeletal age may capture the risk of a future fracture.

4) The discussion should be more balanced, since, apart from a zoledronic acid trial and the quoted metanalysis by Bolland et al. (JCEM 2010;95:1174-81), there is no clear information from randomized controlled trials demonstrating a significant effect of treatments for osteoporosis on mortality. Indeed, as also outlined by Reviewer 2, it seems that the impact of medications affecting bone metabolism on the assessment of skeletal health (osteoporosis drugs, glucocorticoids, PPI, aromatase-inhibitors, etc.) in the Danish National Hospital Discharge Registry was not considered. Moreover, in their conclusive remarks, the Authors propose to replace the fracture probability with skeletal age for individualized fracture risk assessment. Again, given the limited information about the effects of anti-osteoporotic agents on mortality, I'd suggest somehow recommending the "integration" rather than "replacement" of the individual fracture risk assessment with skeletal age, at least at this stage.

*Reviewer #1 (Recommendations for the authors):*

General comments:

The individuals who fracture have more co-morbidities with high mortality risk than the individuals who do not. In order to understand the contribution of a fracture to mortality risk, it would be more appropriate to establish populations of individuals with and without fracture matched with respect to age, gender, and co-morbidities and compare mortality between these groups.

Specific comments:

Results:

l.151-: As the period of fracture identification was from 2001 to 2014, how can the median follow-up be 16 years?

l.163-: Why was the follow-up period for mortality shorter than for fracture? To make the occurrence of fracture and death comparable, I suggest providing both per 100 or 1000 patient-years.

l.166-: It should be clearly stated that the mortality rates in individuals with and without fractures are unadjusted.

Material and methods:

l.320-: Why only data to 2016, this is 6 years ago?

l.330-: How was low-energy fracture determined? Fracture mechanism is usually not captured in registers?

l.343-: Were only comorbidities leading to hospitalisation registered? What about comorbidities diagnosed and treated by GPs?

l.363-: it is not clear how the mortality rate pre-fracture in the patients who fractured was taken into account. The individuals had a higher prevalence of many dead-prone comorbidities and would therefore most likely also have had a higher risk of death before the fracture than the no-fracture population. In addition, how was the risk of a second fracture/recurrent fractures taken into account in the analyses?

Figure 2: the HR was adjusted for the Charlson Co-morbidity index, not specific co-morbidities.

*Reviewer #2 (Recommendations for the authors):*

A few additional points which should be addressed too:

Ascertainment of fractures: The authors should provide some information as to whether these fractures (esp. vertebral fractures) were self-reported or radiographically assessed. In addition to this, was it ascertained that these were fragility fractures?

Patients: Did the authors have any information on the BMD of these subjects? In addition, were medications that affect bone metabolism (osteoporosis drugs, glucocorticoids, PPI, aromatase inhibitors, etc.) documented? If so, why were they not imputed in Cox's proportional hazards model?

---

## [Author Response]

Essential revisions:1) Better clarification of study design and statistical analysis should be required in order to address the conceptual problems raised by Reviewer 1 (e.g. concerning the assessment of comorbidities, or how the information about low-energy fracture obtained from ICD registries, or whether the risk of a second fracture/recurrent fractures was taken into account in the analyses). Additional information is also required about the identification of fractures, particularly vertebral fractures, as outlined by Reviewer 2 (e.g. were all fractures radiographically confirmed?).

Thank you for your suggestions. We have now clarified the study design and statistical analysis.

Comorbidities were ascertained based on the primary diagnosis and all secondary diagnoses of chronic health disorders in the Danish National Hospital Discharge Registry (NHDR) that includes a complete record of diagnoses for all patients treated at Danish hospitals, whether on an outpatient or an inpatient basis. The time frame was within 5 years of the recruitment (i.e., 1996-2000) or of the index fracture. Therefore, comorbidities included not only chronic diseases in the primary diagnosis that require medical attention but also those documented as either secondary diagnoses or at outpatient or emergency visits.

The 5-year window allowed us to capture important chronic diseases. Importantly, the registry-based data of chronic diseases have been found to yield higher positive predictive value than the self-reported ones^1^, indicating the accuracy of the registry-based data in capturing of chronic diseases.

All fractures were ascertained based on the primary diagnosis in the Danish NHDR that required medical attention between 2001 and 2014. All fractures were radiologically confirmed. The fractures included in the current analysis were considered low trauma, as we excluded all high-trauma fractures, including skull, face, nose, finger or toes fractures or those due to traffic accidents.

The nationwide Danish NHDR has included all medical contacts in Denmark since 1995 with excellent completeness of medical records and precision of diagnoses^2,3^. The concordance between fracture reports in the NHDR and patient files was 97%^3^. Importantly, our Danish co-authors (Prof. Bo Abrahamsen, Prof. Peter Vestergaard and Dr. Louise Hansen), who are health-care experts in Denmark confirm that fractures in Denmark are always treated in hospitals, unless they are asymptomatic or overlooked such as some osteoporotic vertebral compressions or small "march" fractures of the smaller bones in the feet which are beyond our analysis scopes. Additionally, previous studies^4^ have shown that self-reported fractures among Danish female nurses were highly concordant, up to 84% for upper hand or wrist fractures and 94% for hip fractures with fractures registered in the NHDR. A sentence for fracture ascertainment has been added:

“All fractures included in the analysis were radiologically ascertained.”

We have then discussed it as a limitation of the study by adding the following sentences to the current manuscript:

“Thirdly, mild, asymptomatic fractures or chronic diseases that did not require medical attention might not be registered in the Danish NHDR. It is nevertheless unlikely that these unregistered fractures and comorbidities would modify the findings significantly, as previous studies have indicated a high concordance between self-reported fractures and those registered in the NHDR^45^ and high positive predictive value of the self-reported comorbidities^44^.”

The analysis did not account for the possibility of subsequent or recurrent fractures, which is indeed beyond the study scope which primarily aimed to quantify the relationship between individual sites of an initial fracture and mortality risk. Mortality risk among patients with a fracture was compared with that of non-fracture individuals, accounting for confounding effect of age and comorbidity severity. Instead, a study that aims to examine the relationship between subsequent or recurrent fractures and mortality should be conducted in a study population of patients who have already sustained an initial fracture. We have added the following sentence to the Statistical analysis section:

“For the first aim, a Cox’s proportional hazards regression was used to quantify the association between an initial specific fracture and mortality, in which both fracture and confounding variables were analysed in a time-dependent manner to minimise the risk of immortal time bias^50^. The analyses did not account for the possibility of subsequent or recurrent fractures as this falls outside of the scope of the study. The models adjusted for confounding effects of age and severity of comorbidities^48^. Age and comorbidities at baseline and those at the time of fracture were also used.”.

2) As underlined by Reviewer 1, the analysis performed did not show that individuals with and without fractures matched for co-morbidities had different mortality risks. Given the large and comprehensive sample, this could be explored in more detail.

We agree that a matched design could be a possible design option. However, such a matched design might be underpowered as it is not always straightforward to obtain sufficient matched pairs to make the planned analysis robust, especially for the analysis that aims to examine mortality risk following each individual fracture site.

We consider that the multivariable model that we used is methodologically rigorous and statistically equivalent to the matched design in the control for confounding effects. Specifically, the multivariable-adjusted Cox’s proportional hazard regression used in this study is capable of accounting for the confounding effect of age and comorbidities as a matched design can do. Moreover, the adjusted analysis allows us to make use of data of all participants in the study population, ultimately improving the analysis power and robustness that a matched design might not be able to achieve.

3) Additional information is required to better explain how the assessment of skeletal age may capture the risk of a future fracture.

Thank you for your suggestion. Traditional fracture risk assessment tools (i.e., FRAX, Garvan or QFracture)^5^ provide an estimated absolute risk of fracture over a period of time. By contrast, skeletal age reflects the excess mortality as a consequence of a fracture for an individual with a specific risk profile. For instance, a 60-year man with a hip fracture is estimated to the skeletal age of a 66-year old, indicating a 6-year loss of life. This operational definition is consistent with the definition of 'Effective Age' in the medical literature^6^.

We have added more details to explain how the assessment of skeletal age would provide the conceptual risk of both fracture and post-fracture mortality as follows:

“Unlike the current fracture risk assessment tools^17^ which estimate the probability of fracture over a period of time using probability-based metrics, such as relative risk and absolute risk, skeletal age quantifies the consequence of a fracture using a natural frequency metric. A natural frequency metric has been consistently shown to be easier and more friendly to doctors and patients than the probability-based metrics^9 11 30^. It is not straightforward to appreciate the importance of the two-fold increased risk of death (i.e., relative risk = 2.0) without knowing the background risk (i.e., 2 folds of 1% would remarkably differ from 2 folds of 10%). By contrast, for the same 2-fold mortality risk of hip fracture, telling a 60-year man with a hip fracture that his skeletal age would be 66 years old, equivalent to a 6-year loss of life, is more intuitive. The skeletal age can also be interpreted as the individual being in the same risk category as a 66-year-old with 'favorable risk factors' or at least the ones that are potentially modifiable. Hence, an older skeletal age means a greater risk of fracture.”.

4) The discussion should be more balanced, since, apart from a zoledronic acid trial and the quoted metanalysis by Bolland et al. (JCEM 2010;95:1174-81), there is no clear information from randomized controlled trials demonstrating a significant effect of treatments for osteoporosis on mortality.

Thank you for your suggestion. We agree that the effect of osteoporosis treatments on mortality remains contentious, but this is not our primary focus. Our focus is to demonstrate the use of skeletal age to convey a positive impact of a treatment, and the mention of zoledronic acid's effect was simply for illustration purpose. We have decided to remove the section related to the benefit of pharmacological treatment on post-fracture mortality.

Indeed, as also outlined by Reviewer 2, it seems that the impact of medications affecting bone metabolism on the assessment of skeletal health (osteoporosis drugs, glucocorticoids, PPI, aromatase-inhibitors, etc.) in the Danish National Hospital Discharge Registry was not considered.

Our analysis could not factor in the impact of medications that impact bone metabolism. Nevertheless, these medications unlikely have a substantially effect on the risk of post-fracture mortality. First, only 0.3% of men and 2.2% of women aged 50+ in Denmark were prescribed bisphosphonate, and bisphosphonate or raloxifene, respectively in 1999^7^, while only 4.1% of men and 9.2% of women with hip fracture began bisphosphonates in 2004^8^. Secondly, the most recent meta-analysis of 38 randomised clinical trials of drug therapies that included 101,642 patients with osteoporosis suggested these bone metabolism-affecting medications were not associated with improved survival (RR: 0.98; 95% CI: 0.91 to 1.05; I^2^= 0%)^9^. Finally, it is not ideal to examine the impact of a pharmacological intervention using our observational data which are inevitably confounded by uncontrolled biases and residual confounding effects.

We have acknowledged it as a study limitation as follows:

“Finally, our analysis did not include the examination of the effect of bone metabolism-affecting medications on the assessment of skeletal health. However, these medications were prescribed to only a few individuals aged 50 and over in Denmark^46^, even for those with hip fractures^47^, and there is no substantial evidence from randomized controlled trials of their impact on post-fracture mortality^6^.”

Moreover, in their conclusive remarks, the Authors propose to replace the fracture probability with skeletal age for individualized fracture risk assessment. Again, given the limited information about the effects of anti-osteoporotic agents on mortality, I'd suggest somehow recommending the "integration" rather than "replacement" of the individual fracture risk assessment with skeletal age, at least at this stage.

We agree with the reviewer that the assessment of skeletal age should not 'replace' the current fracture risk assessment. We have removed the word 'replace' from the conclusion statement.

Reviewer #1 (Recommendations for the authors):General comments:The individuals who fracture have more co-morbidities with high mortality risk than the individuals who do not. In order to understand the contribution of a fracture to mortality risk, it would be more appropriate to establish populations of individuals with and without fracture matched with respect to age, gender, and co-morbidities and compare mortality between these groups.

We have commented on the issue of matching *versus* multivariable regression model. Both approaches are designed to control for the effect of potential confounders, but each approach has strengths and limitations, with the matching design often having lower power than the regression adjustment. Some studies have shown that Cox’s regression model applied to the entire cohort is statically more powerful than a matched design. In our study, we chose Cox’s model to factor in the effects of age, gender, and comorbidities on the fracture-mortality association. This analytic strategy also allowed us to estimate the skeletal age for each individual based on the individual’s comorbidity profile.

Specific comments:Results:l.151-: As the period of fracture identification was from 2001 to 2014, how can the median follow-up be 16 years?

Thank you for pointing it out. The median follow-up was 16 years for the whole study period (from 2001 to 2016), but 14 years for fracture identification (from 2001 to 2014). We have corrected this typo as:

“During a median of 14 years of follow up (interquartile range [IQR]: 6.5, 14.0 years)”.

l.163-: Why was the follow-up period for mortality shorter than for fracture? To make the occurrence of fracture and death comparable, I suggest providing both per 100 or 1000 patient-years.

The follow-up time for mortality was shorter for fracture patients who were more likely to die sooner following a fracture than those who did not sustain a fracture. We reported the mortality rate as the number of deaths per 100 person-years for both fracture and non-fracture groups.

In our analysis, the contribution of initial fractures to mortality was examined in a time-dependent analysis in which both the occurrence of fracture (as the main exposure) and the confounding effect of age and comorbidities (as the predefined covariates) were analysed in a time-dependent manner to minimise the risk of immortal time bias^21^. Statistically, the follow-up time for fracture patient was split into the pre- and post-fracture periods. The pre-fracture period (i.e., the follow-up time between the study entry and the date of fracture for those with an incident fracture) is considered unexposed time, while its clustering effect is accounted for in a Cox’s proportional hazards model^21^.

We have added details and a reference to explain the concept of time-dependent analysis further as follows:

“By contrast, the follow-up time for those with an incident fracture was split into the pre-fracture (i.e., unexposed) and post-fracture (i.e., exposed) period for which their clustering effect was accounted for in the Cox’s proportional hazards regression. We conducted a time-dependent analysis to quantify the relationship between an incident exposure (i.e., an incident fracture) and the outcome of interest (i.e., post-fracture mortality) and to minimize the risk of immortal time bias^50^.”

l.166-: It should be clearly stated that the mortality rates in individuals with and without fractures are unadjusted.

Thank you. We have revised the text to indicate that the mortality rates were unadjusted.

Material and methods:l.320-: Why only data to 2016, this is 6 years ago?

This study is a part of our collaborative project (FCEK 706667) that was approved in 2019 using medical records between 1996 and 2016 in the Danish NHDR. There is usually a time gap between the actual data and when the data are linked and available for a research project.

l.330-: How was low-energy fracture determined? Fracture mechanism is usually not captured in registers?

Please refer to the above response for more detail. Briefly, we used the ICD-10 codes in the primary analysis for any healthcare contact from the Danish NHDR to capture all fractures among individuals aged 50+ years that require medical attention between 2001 and 2014. Fractures in Denmark are always treated in hospitals, unless they are asymptomatic or overlooked such as some osteoporotic vertebral compressions or small "march" fractures of the smaller bones in the feet which are beyond our analysis scopes. All fractures included in the analysis were radiologically confirmed. We excluded high-trauma fractures, including skull, face, nose, finger or toes fractures or those due to traffic accidents. We have additionally acknowledged the possibilities, though unlikely that the analysis might have missed mild, asymptomatic fractures that did not receive medical attention.

l.343-: Were only comorbidities leading to hospitalisation registered? What about comorbidities diagnosed and treated by GPs?

No, comorbidities were identified using not only the primary diagnosis that requires medical attention but also all secondary diagnoses in any healthcare contact in both inpatient and outpatient visits. Unfortunately, the Danish NHDR does not include comorbidities diagnosed and treated by GPs. However, given the high concordance between the self-reported comorbidities and those in the registry^1^, it is unlikely that the current analysis might have missed substantial number of comorbidities. We have additionally acknowledged it as a study limitation that mild comorbidities that did not require medical attention might have not been registered in the Danish NHDR.

l.363-: it is not clear how the mortality rate pre-fracture in the patients who fractured was taken into account. The individuals had a higher prevalence of many dead-prone comorbidities and would therefore most likely also have had a higher risk of death before the fracture than the no-fracture population.

For individuals who sustained an incident fracture during the follow-up period, the follow-up time was split into two time periods, i.e., the pre-fracture (~pre-exposure) and post-fracture (post-exposure) periods to minimise the risk of immortal time bias^16^. As a result, no death event, as the primary endpoint of interest occurs during the pre-exposure period of the fracture patients since they have to survive to sustain a fracture. As mentioned earlier, we have added sentences and a reference to elaborate the concept of a time-dependent analysis for controlling immortal time bias.

The issue of competing risk of death, i.e., some participants might have died before they had a chance to sustain a fracture, is not a statistical issue in the current analysis that models all-cause death as the primary endpoint.

In addition, how was the risk of a second fracture/recurrent fractures taken into account in the analyses?

As explained earlier, the current analysis did not account for the possibility of subsequent or recurrent fractures which falls outside of the scope of the study.

Figure 2: the HR was adjusted for the Charlson Co-morbidity index, not specific co-morbidities.

Thank you. We have corrected the sentence as suggested.

Reviewer #2 (Recommendations for the authors):A few additional points which should be addressed too:Ascertainment of fractures: The authors should provide some information as to whether these fractures (esp. vertebral fractures) were self-reported or radiographically assessed. In addition to this, was it ascertained that these were fragility fractures?

Please refer to the above comment for details. Briefly, we only included the fractures that required medical attention and were radiologically ascertained. We excluded high-trauma fractures among participants aged 50+ years.

Patients: Did the authors have any information on the BMD of these subjects?

Unfortunately, we did not have any BMD data of the participants.

In addition, were medications that affect bone metabolism (osteoporosis drugs, glucocorticoids, PPI, aromatase inhibitors, etc.) documented? If so, why were they not imputed in Cox's proportional hazards model?

As clarified above, we did not consider the effect of medication in our analysis, and we acknowledge that this is a potential weakness. Nevertheless, it is very unlikely that these medications would have a substantial impact on the risk of post-fracture mortality.

References:

1. Lujic S, Simpson JM, Zwar N, Hosseinzadeh H, Jorm L. Multimorbidity in Australia: Comparing estimates derived using administrative data sources and survey data. *PloS one* 2017; 12(8): e0183817.

2. Andersen TF, Madsen M, Jorgensen J, Mellemkjoer L, Olsen JH. The Danish National Hospital Register. A valuable source of data for modern health sciences. *Dan Med Bull* 1999; 46(3): 263-8.

3. Vestergaard P, Mosekilde L. Fracture risk in patients with celiac Disease, Crohn's disease, and ulcerative colitis: a nationwide follow-up study of 16,416 patients in Denmark. *Am J Epidemiol* 2002; 156(1): 1-10.

4. Hundrup YA, Hoidrup S, Obel EB, Rasmussen NK. The validity of self-reported fractures among Danish female nurses: comparison with fractures registered in the Danish National Hospital Register. *Scand J Public Health* 2004; 32(2): 136-43.

5. Beaudoin C, Moore L, Gagne M, et al. Performance of predictive tools to identify individuals at risk of non-traumatic fracture: a systematic review, meta-analysis, and meta-regression. *Osteoporos Int* 2019; 30(4): 721-40.

6. Spiegelhalter D. How old are you, really? Communicating chronic risk through 'effective age' of your body and organs. *BMC Med Inform Decis Mak* 2016; 16: 104.

7. Vestergaard P, Rejnmark L, Mosekilde L. Osteoporosis is markedly underdiagnosed: a nationwide study from Denmark. *Osteoporos Int* 2005; 16(2): 134-41.

8. Roerholt C, Eiken P, Abrahamsen B. Initiation of anti-osteoporotic therapy in patients with recent fractures: a nationwide analysis of prescription rates and persistence. *Osteoporos Int* 2009; 20(2): 299-307.

9. Cummings SR, Lui LY, Eastell R, Allen IE. Association Between Drug Treatments for Patients With Osteoporosis and Overall Mortality Rates: A Meta-analysis. *JAMA Int Med* 2019; 179(11): 1491-500.

10. Chen W, Simpson JM, March LM, et al. Comorbidities Only Account for a Small Proportion of Excess Mortality After Fracture: A Record Linkage Study of Individual Fracture Types. *J Bone Miner Res* 2018; 33(5):795-802

11. Vestergaard P, Rejnmark L, Mosekilde L. Increased mortality in patients with a hip fracture-effect of pre-morbid conditions and post-fracture complications. *Osteoporos Int* 2007; 18(12): 1583-93.

12. Tran T, Bliuc D, Hansen L, et al. Persistence of Excess Mortality Following Individual Nonhip Fractures: A Relative Survival Analysis. *J Clin Endocrinol Metab* 2018; 103(9): 3205-14.

13. Tran T, Bliuc D, Ho-Le T, et al. Association of Multimorbidity and Excess Mortality After Fractures Among Danish Adults. *JAMA Netw Open* 2022; 5(10): e2235856.

14. Henderson R, Keiding N. Individual survival time prediction using statistical models. *J Med Ethics* 2005; 31(12): 703-6.

15. Kulinskaya E, Gitsels LA, Bakbergenuly I, Wright N. Calculation of changes in life expectancy based on proportional hazards model of an intervention. *Insur Math Econ* 2020; 93: 27-35.

16 Lopez-Gonzalez AA, Aguilo A, Frontera M, et al. Effectiveness of the Heart Age tool for improving modifiable cardiovascular risk factors in a Southern European population: a randomized trial. *Eur J Prev Cardiol* 2015; 22(3): 389-96.

17. Lyles KW, Colon-Emeric CS, Magaziner JS, et al. Zoledronic acid and clinical fractures and mortality after hip fracture. *N Engl J Med* 2007; 357(18): 1799-809.

18. Reid IR, Horne AM, Mihov B, et al. Fracture Prevention with Zoledronate in Older Women with Osteopenia. *N Engl J Med* 2018; 379(25): 2407-16.

19. Bonner C, Batcup C, Cornell S, et al. Interventions Using Heart Age for Cardiovascular Disease Risk Communication: Systematic Review of Psychological, Behavioral, and Clinical Effects. *JMIR Cardio* 2021; 5(2): e31056.

20. Svendsen K, Jacobs DR, Morch-Reiersen LT, et al. Evaluating the use of the heart age tool in community pharmacies: a 4-week cluster-randomized controlled trial. *Eur J Public Health* 2020; 30(6): 1139-45.

21. Suissa S. Immortal time bias in pharmaco-epidemiology. *Am J Epidemiol* 2008; 167(4): 492-9.